# Impact of Hospitalization on Sarcopenia, NADPH-Oxidase 2, Oxidative Stress, and Low-Grade Endotoxemia in Elderly Patients

**DOI:** 10.3390/antiox14030304

**Published:** 2025-03-01

**Authors:** Chiara Bagnato, Arianna Magna, Elena Mereu, Sciaila Bernardini, Simona Bartimoccia, Roberta Marti, Pietro Enea Lazzerini, Alessandra D’Amico, Evaristo Ettorre, Giovambattista Desideri, Pasquale Pignatelli, Francesco Violi, Roberto Carnevale, Lorenzo Loffredo

**Affiliations:** 1Department of Clinical Internal, Anaesthesiologic and Cardiovascular Science, Sapienza University of Rome, Viale del Policlinico, 155, 00161 Rome, Italy; chiara.bagnato@uniroma1.it (C.B.); arianna.magna@uniroma1.it (A.M.); elena.mereu@uniroma1.it (E.M.); simona.bartimoccia@uniroma1.it (S.B.); roberta.marti@uniroma1.it (R.M.); evaristo.ettorre@uniroma1.it (E.E.); giovambattista.desideri@uniroma1.it (G.D.); pasquale.pignatelli@uniroma1.it (P.P.); 2Department of Medical Sciences, Surgery and Neurosciences, University of Siena, 53100 Siena, Italypietro.lazzerini@unisi.it (P.E.L.); 3Department of Medical and Surgical Sciences and Biotechnologies, Sapienza University of Rome, 00185 Rome, Italy; alessandra.damico@uniroma1.it; 4IRCCS Neuromed, 86077 Pozzilli, Italy; roberto.carnevale@uniroma1.it; 5Sapienza University of Rome, 00185 Rome, Italy; francesco.violi@uniroma1.it; 6Department of Medical-Surgical Sciences and Biotechnologies, Sapienza University of Rome, 04100 Latina, Italy

**Keywords:** sarcopenia, oxidative stress, NADPH oxidase, NOX2, NADPH oxidase, muscle ultrasound, hospitalization, aging, physical activity

## Abstract

Background: Hospitalization in older adults often worsens sarcopenia due to prolonged bed rest, poor nutrition, and inactivity. This study examined how hospitalization impacts muscle mass, focusing on oxidative stress and gut-derived endotoxemia. Methods: Thirty-one hospitalized older adults were compared with 31 outpatients. Ultrasound was used to measure the thickness of the rectus femoris (RF), intercostal, and diaphragmatic muscles at admission and discharge. Serum levels of LPS, zonulin, sNOX2-dp, and H_2_O_2_ were also assessed. Results: Hospitalized patients had higher serum levels of sNOX2-dp, H_2_O_2_, LPS, and zonulin than outpatients. In hospitalized patients, significant increases were observed at discharge compared to admission levels in sNOX2-dp (20.9 ± 6.5 to 23.8 ± 7.5 pg/mL; *p* = 0.004), H_2_O_2_ (24.4 ± 9.8 to 32.8 ± 14.5 µM; *p* = 0.01), LPS (30.4 ± 12.6 to 43.3 ± 16.35 pg/mL; *p* < 0.001), and zonulin (2.06 ± 1.23 to 2.95 ± 1.33 ng/mL; *p* < 0.001). Ultrasound data revealed a reduction in RF muscle thickness (−35%) (0.58 ± 0.29 to 0.38 ± 0.31 cm, *p* < 0.001), intercostal muscle thickness (−28%) (0.22 ± 0.08 to 0.16 ± 0.06 cm, *p* < 0.001), and diaphragmatic muscle thickness (−26%) (0.19 ± 0.06 to 0.14 ± 0.04 cm, *p* < 0.001) at discharge compared to admission. Additionally, muscle strength, measured using the hand-grip test, showed a 25% reduction. Regression analysis revealed correlations between RF muscle loss and increases in sNOX2-dp and H_2_O_2_, as well as between NOX2, H_2_O_2_, and LPS with zonulin. Conclusions: Hospitalization in older adult patients elevates NOX2 blood levels, correlating with reduced muscle mass. Increased low-grade endotoxemia may trigger NOX2 activation, generating oxidative stress that accelerates muscle degeneration and can lead to sarcopenia.

## 1. Introduction

Sarcopenia is characterized by the progressive loss of skeletal muscle mass and strength associated with a deterioration in performance [1]. In older adult patients, sarcopenia constitutes a significant social and health problem, as it increases frailty, disability, infections, bedsores, the risk of falls, and bed rest [2]. Hospitalization seems to further promote the decline in muscle mass and strength and is one of the main causes of sarcopenia, especially in older adults [1]. Prolonged bed rest, inadequate nutrition, and acute stress due to the disease can, in fact, accelerate sarcopenia, making functional recovery more difficult and prolonging hospital stay [1]. However, a quantitative measure of the actual and real damage produced by hospitalization on the loss of muscle thickness and strength is not fully known.

Probably, the reason for this lies in the lack of tools to intercept and follow-up sarcopenia. In fact, currently muscle investigation techniques are mainly based on magnetic resonance imaging (MRI) and computed tomography (CT), which are very complicated to reproduce in routine hospital clinical practice and impossible at the patient’s bedside. In this complex panorama, muscle ultrasound stands out as a valid technique for making an early diagnosis and monitoring the progression of sarcopenia during the hospital visit [3].

Sarcopenia has a multifactorial etiology, influenced by genetic, lifestyle, and environmental factors. It is a complex metabolic syndrome and causes a state of severe muscle and systemic wasting due to chronic inflammatory processes such as cachexia, the reduction of anabolic and hormonal triggers, increased catabolism, and insulin resistance [4]. Recent research also highlights the critical roles of oxidative stress and low-grade endotoxemia in the development and progression of sarcopenia, especially during hospitalization in older adults [5]. In skeletal muscle, reactive oxygen species (ROS) have a dual function: at low levels, they increase muscle strength and adaptation to exercise, while at high levels, they lead to a decrease in muscle performance [6,7]. In fact, under normal physiological conditions, ROS play a key role in cell signaling and homeostasis [6,7]. However, in older adults, oxidative stress results from a decrease in the activity and quantity of mitochondria, from reduced bioenergetic production in the muscle with an imbalance between the production of reactive oxygen species (ROS) and the antioxidant defenses of the organism [6,7]. An accumulation of ROS produced by the endoplasmic reticulum (ER) can also damage cellular components, including proteins, lipids, and DNA, leading to the dysfunction and death of muscle cells. In particular, ER stress can inhibit the mTORC1 complex that mediates the response to the anabolic stimulus of nutrients and contractile activity, thus inducing a reduced regenerative capacity of the muscle detectable during aging [8,9].

Aging is associated with increased oxidative stress due to chronic low-grade inflammation, often caused by the translocation of bacterial-derived proteins such as lipopolysaccharides (LPS) from the gut to the bloodstream due to impaired intestinal tight junctions caused by altered zonulin [10,11]. In vitro studies have shown that LPS increases muscle catabolism through the activation of toll-like receptor-4 (TLR-4), which triggers transcription factors such as ubiquitin proteasome and NF-kb, which in turn release IL-6, IL-1, CRP, and TNF alpha [12,13]. These circulating cytokines contribute to muscle protein degradation. Therefore, it is hypothesized that sarcopenia in older adults is associated with increased TLR4 expression/signaling, which may be secondary to metabolic endotoxemia, as it is activated by LPS [14]. In skeletal muscle, the primary source of ROS during normal function appears to be NADPH oxidase 2 (NOX2), which is also a major source of superoxide anion in humans [15]. Low levels of endotoxemia via LPS may act as a trigger for NOX2 activation, contributing to the cascade of oxidative stress and inflammation via cytokine synthesis described above [16].

Recent studies have shown that the composition of microorganisms residing in the digestive system flora varies with increasing age [10]. With aging, older adults suffer from increased intestinal mucosal permeability, such that low-grade endotoxin markers can enter the bloodstream and cause chronic inflammation. Changes in the senescent intestinal bacterial population could alter the absorption of nutrients and energy substrates, such as proteins, lipids, and glucose, promoting cachexia and malnutrition, factors that determine poor systemic and especially muscular health [10].

Frailty and sarcopenia are conditions that frequently coexist in the same patient and impact multiple domains, such as physical performance, cognitive reserve, autonomy, mood swings, and the nutritional sphere. Older adults affected by sarcopenia are more frail. In fact, according to Bortz’s “vicious circle of sarcopenia”, older adults with sarcopenia will increasingly avoid physical effort, preferring a sedentary lifestyle, worsening physical performance, daily autonomy, cognitive capacity and mood, nutritional status, and cardiovascular function, increasing mortality outcomes. Hospitalization could exacerbate this vicious circle [17].

During hospitalization, factors such as acute illnesses, surgeries, and the body’s inflammatory response can further elevate oxidative stress levels [18,19]. This heightened oxidative environment accelerates muscle loss, complicating the management of sarcopenia and increasing the likelihood of negative outcomes [20].

The aim of this study was to evaluate the effect of hospitalization on muscle mass (assessed by ultrasound measurement of muscle thickness) as well as its association with oxidative stress (measured by blood levels of NOX2 and H_2_O_2_) and low-grade endotoxemia (evaluated by blood levels of LPS and zonulin).

## 2. Materials and Methods

We conducted a study on a sample of 31 hospitalized patients aged 65 years and older, comparing them to 31 age- and gender-matched outpatients. The inclusion criteria were a hospital stay of more than 5 days and an SARC-F Questionnaire score of 4 or higher, which is predictive of sarcopenia [21]. The exclusion criteria were a hospital stay of less than 5 days, a SARC-F questionnaire score of less than 4, and severe sepsis. The data collected included medical history; anthropometric and dynamometric measurements (hand-grip test) [22]; and the administration of the Barthel Index [23], the Mini Nutritional Assessment (MNA) [24] to assess the risk of malnutrition, the Mini-Mental State Examination (MMSE) [25], the Geriatric Depression Scale (GDS) [26] at admission and discharge; and the Cumulative Illness Rating Scale (CIRS) [27,28]. Additionally, muscle ultrasound was performed on the rectus femoris (RF) muscle, intercostal muscle, and diaphragm at both admission and discharge.

Muscle ultrasound was employed to study body composition, as it is a reliable and valid method for quantifying tissue thickness. With advancing age, healthy muscle tissue becomes more echogenic due to the progressive replacement of muscle by adipose and fibrous tissue, and muscle thickness tends to decrease. The muscle ultrasound was performed in B-mode using a 5–7.5 MHz linear probe (Samsung V5 ultrasound) at the medial portion of the rectus femoris muscle, as previously described [29]. The patient was positioned supine, without any recent muscle exertion (within the preceding 30–60 min), with both knees extended, relaxed, and the toes pointing toward the ceiling. The probe was positioned longitudinally along the long axis of the muscle fascia, with an adequate amount of gel applied, and minimal pressure was exerted to avoid excessive compression of the muscle. The anatomical structures observed from the top to the bottom of the quadriceps femoris scan included the skin, subcutaneous layer, rectus femoris muscle, vastus intermedius muscle, and femoral head. For each scan, three consecutive measurements of the rectus femoris muscle thickness were taken, from which the mean value in centimeters (cm) and the standard deviation were calculated.

In accordance with Rustani K et al. [29], we used 0.9 cm as the threshold value for RF thickness to distinguish healthy individuals from those with sarcopenia, with a sensitivity of 100%, a specificity of 64%, and a negative predictive value of 100%.

In a subgroup of recruited patients, muscle ultrasound of the parasternal intercostal respiratory muscles and the diaphragm was also performed as previously described [30]. The patient was placed in a supine position with the trunk inclined at a 45° angle. For visualization of the parasternal muscles, a 5–7.5 MHz linear probe was positioned at the level of the second intercostal space, 6–8 cm from the sternal margin, where the thickness of the intercostal muscles was measured in B-mode and M-mode at the inspiratory peak and in the tele-expiratory phase. For visualization of the diaphragm, the probe was placed at the level of the 8th intercostal space on the right mid-axillary line. A 3–5 MHz convex probe was used to measure diaphragmatic excursion in M-mode, and a 5–7.5 MHz linear probe was used to measure muscle thickness in B-mode and M-mode at peak inspiration and in the tele-expiratory phase. All measurements were performed by the same healthcare professionals to reduce inter-individual variability. Muscle ultrasound assessments were conducted in a blinded manner; the ultrasound operator was unaware of the patient’s status.

We also performed a comprehensive geriatric assessment on admission and discharge, including data from the SARC-F [18], the Barthel Index [20], the Mini Nutritional Assessment (MNA) [21] to assess malnutrition risk, the Mini-Mental State Examination (MMSE) [22], the Geriatric Depression Scale (GDS) [23], and the Cumulative Illness Rating Scale (CIRS) [24,25]. The assessment of muscle function using the SARC-F questionnaire is only part of the sarcopenia screening, where we examine the number of falls, the ability to lift weights, the type of gait and trajectory in walking and/or the need for assistance, and the ability to climb stairs and to get up from a chair. However, in accordance with the literature, we performed the combination of SARC-F + anthropometric measures, such as calf circumference and arm circumference, as they significantly improve the sarcopenia screening performance of SARC-F, allowing its use in clinical practice [31]. The Barthel Index is a tool to assess functional status in daily activities by investigating aspects such as feeding, personal hygiene, dressing, bowel control, bladder control, bathroom use, transfers (from bed to chair), mobility (on flat surfaces), and stairs. With a score of 0 to 100, representing the highest level of autonomy, we obtain valuable information on the degree of the patient’s disability [32]. Nutrition is a key factor in sarcopenia, and dietary intake has also been assessed using the MNA. The MNA allows for rapid nutritional assessment and was developed to evaluate malnutrition status as part of the standard assessment of patients who are older or frail in clinics, nursing homes, and hospitals. The MNA shows a statistically significant correlation with morbidity and mortality. After accurately completing the screening section, the scores are added together to obtain the total score. If the result is 11 or lower, the next global assessment is necessary, as it indicates a high-risk condition for malnutrition [33]. The Mini-Mental State Examination (MMSE) is one of the most commonly used scales to define the degree of cognitive impairment and monitor its progression across 6 distinct domains. The scores obtained in the investigated domains (Constructive Ability, Orientation, Recall, Language, Attention and Computation, Recording), if impaired, indicate the brain areas affected by neurodegeneration and must be corrected for the patient’s level of education [34]. The Geriatric Depression Scale (GDS), which specifically assesses the degree of depression in older adults, was tested for its reliability and validity. The test questions identify individuals as euthymic, mildly depressed, or severely depressed on the basis of the Research Diagnostic Criteria (RDC) for depression [26]. The Cumulative Illness Rating Scale (CIRS) is a tool that aims to summarize the overall severity of a disease on the basis of clinical information and its progressive worsening in seriousness from the time of admission to the time of discharge with an increasing score [35].

### 2.1. sNOX2-dp Assay

NOX2 activation was measured using an ELISA method to detect soluble NOX2-derived peptide (sNOX2-dp), as previously described [36]. Briefly, this method involves recognizing the peptide through binding to a specific monoclonal antibody targeting the amino acid sequence (224–268), which corresponds to the extracellular domain of NOX2. Results were expressed in pg/mL, with intra-assay and inter-assay coefficients of variation of 8.95% and 9.01%, respectively.

### 2.2. Hydrogen Peroxide (H_2_O_2_) Production

Hydrogen peroxide (H_2_O_2_) concentrations were determined using a colorimetric assay according to the manufacturer’s instructions (Abcam, Cambridge, UK). Results were expressed in micromoles (μM). The intra-assay and inter-assay coefficients of variation were both <10%.

### 2.3. Serum Zonulin

Serum zonulin levels were measured using a commercially available ELISA kit (Elabscience). A microplate pre-coated with an antibody specific to zonulin was used. Standards and 100 μL of patient serum samples were added to the plate and incubated for 90 min at 37 °C. Subsequently, a biotinylated detection antibody specific for zonulin and an Avidin-Horseradish Peroxidase (HRP) conjugate were added. Results were expressed in ng/mL, with intra-assay and inter-assay coefficients of variation remaining within 10%.

### 2.4. Serum LPS Assay

Serum LPS levels were determined using a commercial ELISA kit (Cusabio, Wuhan, China). Standards and samples were plated onto a microplate pre-coated with an antibody specific for LPS and incubated for 2 h at room temperature. Following incubation, the samples were read at 450 nm. Results were expressed in pg/mL, with intra-assay and inter-assay coefficients of variation below 10%.

The experimental procedure was approved by the Institutional Review Board at Sapienza University of Rome (ref. no. 7088/23) and was conducted in accordance with the Declaration of Helsinki.

### 2.5. Sample Size Determination

The minimum sample size was calculated for a two-tailed, one-sample Student’s *t*-test.

Based on data from a previous pilot study 60 patients were required to achieve a 95% chance of detecting a significant difference at the 5% significance level. The expected difference in LPS levels between the control outpatient group (12 pg/mL) and the hospitalized group (25 pg/mL), with a standard deviation of 15, was used for this calculation.

The number of necessary hospitalized patients was estimated under the assumption that the mean LPS would be 30.0 and 40.0 pg/mL at admission and discharge, respectively, with a common standard deviation of 15.0. With a type 1 risk of 5.0% and a type 2 risk of 10.0%, a bilateral test, and a dropout rate estimated at 10.0%, we estimated that at least 30 patients would be necessary for this study.

### 2.6. Statistical Analysis

Statistical analyses were performed using SPSS 25.0 software for Windows (SPSS, Chicago, IL, USA). The Shapiro–Wilk test was employed to assess the normality of variable distributions. Data are presented as means ± standard deviations (SDs). Differences between groups were analyzed using analysis of variance (ANOVA) for normally distributed data or the Mann–Whitney U test for non-normally distributed data. The Wilcoxon test and paired Student’s *t*-test were used to compare the two study phases (hospital admission and discharge). All analyses performed with parametric tests were replicated using non-parametric tests. Bivariate correlations were assessed using Spearman’s rank correlation test.

## 3. Results

Thirty-one hospitalized patients and outpatients were consecutively recruited. The clinical characteristics of this population are described in Table 1.

Compared to outpatients, hospitalized patients exhibited higher levels of sNOX2-dp (9.4 ± 5.6 vs. 20.9 ± 6.5; *p* < 0.001), H_2_O_2_ (9.3 ± 3.0 vs. 24.4 ± 9.8; *p* < 0.001), LPS (11.2 ± 3.2 vs. 30.4 ± 12.6; *p* < 0.001), and zonulin (1.5 ± 0.43 vs. 2.06 ± 1.23; *p* = 0.007) (Table 1). Additionally, hospitalized individuals had a higher prevalence of diabetes mellitus, dyslipidemia, dementia, heart failure, and a history of smoking, and they were also leaner (Table 1).

Compared to the ultrasound data obtained at admission, a statistically significant reduction was observed at discharge in the RF muscle (−35%) (0.58 ± 0.29 cm vs. 0.38 ± 0.31 cm, *p* < 0.001) (Figure 1A), in the intercostal muscle (−28%) (0.22 ± 0.08 cm vs. 0.16 ± 0.06 cm, *p* < 0.001), and in the diaphragmatic muscle (−26%) (0.19 ± 0.06 cm vs. 0.14 ± 0.04 cm, *p* < 0.001).

A reduction in muscle strength (−25%) measured using the hand-grip test was observed in this population (Table 2).

Compared to admission, significant increases in sNOX2-dp (20.9 ± 6.5 vs. 23.8 ± 7.5; *p* = 0.004), serum levels of H_2_O_2_ (24.4 ± 9.8 vs. 32.8 ± 14.5; *p* = 0.01), LPS (30.4 ± 12.6 vs. 43.3 ± 16.35; *p* < 0.001), and zonulin (2.06 ± 1.23 vs. 2.95 ± 1.33; *p* < 0.001) were observed at the end of hospitalization (Table 2 and Figure 1C–E). Additionally, a significant increase in the SARC-F, CIRS, and GDS scores and a decrease in the Barthel Index, MMSE, and MNA scores were observed in this study (Table 2).

A simple linear regression analysis revealed a significant correlation between the deltas (% of differences between the values measured at the beginning and end of hospitalization) of RF and NOX2 (R: −0.388; *p* = 0.031) and H_2_O_2_ (R: −0.364; *p* = 0.044), while NOX2 variations also correlated with H_2_O_2_ (R: 0.252; *p* = 0.171) and LPS with zonulin (R: 0.537; *p* = 0.002).

## 4. Discussion

The significant impact of hospitalization on sarcopenia in older adult patients is central to our study. Moreover, elevated oxidative stress and low-grade endotoxemia are associated with the progression of sarcopenia during hospitalization.

The development and worsening of sarcopenia, mainly due to reduced mobility or bed rest in older adult patients during hospitalization, is well-documented in the literature [37,38].

Older adult patients who become sarcopenic or whose pre-existing sarcopenia worsens during hospitalization are unlikely to recover their physical condition and face an increased risk of complications, such as falls, infections, hospitalization, and death [1]. This condition will further exacerbate healthcare costs [39].

Hospitalization-induced sarcopenia may be responsible for the increased muscle catabolism, which, as evidenced by previous studies, is predominantly mediated by NADPH oxidase type 2 [9,15]. This enzyme is involved in the production of ROS in numerous processes in muscle fibers, including ongoing metabolism during exercise, gene regulation, insulin metabolism, and muscle mass regulation [40]. Animal studies have shown that NOX2 plays a fundamental role in anabolic resistance, a process that contributes to reduced muscle protein production in older adults [40]. Studies on murine models have shown that mice with a genetic deficiency of NOX2 exhibit greater exercise-induced muscle hypertrophy compared to controls [41]. The increased oxidative stress generated by NOX2 may itself be a factor that worsens muscle degradation and sarcopenia in older adult patients [40]. Physical activity also appears to promote the preservation of the number and functionality of mitochondria in skeletal muscle, and it is indeed considered one of the most important determinants of mitochondrial functionality in old age [40]. Furthermore, the increase in ROS and RNS levels further promotes mitochondrial damage, generating a self-sustaining and amplified damage mechanism in older adults [40].

Hospitalization in older adults is associated with an increase in intestinal permeability mediated by zonulin that alters intestinal tight junctions and releases serum LPS into the bloodstream, which acts as a trigger for the activation of NADPH oxidase type 2, which self-maintains this vicious circle. Increased low-grade endotoxemia can lead to chronic inflammation through the activation of the TLR-4 receptor, which promotes the transcription of proinflammatory cytokines and is potentially detrimental to muscle health [16]. In conditions such as hospitalization and especially with aging, the levels of anabolic factors such as thyroid hormones, IGF-1, GH, and testosterone are reduced, causing a decrease in muscle protein synthesis and an increase in muscle degradation, related to a reduction in mass and strength. In fact, only resistance exercise seems to be able to reactivate this balance in favor of anabolism and should be considered a mobilization strategy from the first moments of hospitalization [42].

Aging appears to be associated with a decline in microbial diversity and an increase in potentially harmful bacteria (such as Bacteroides), contributing to dysbiosis [43]. Protein turnover in muscle tissue appears to vary due to alterations in intestinal mucosal permeability and the immune system with the development of less microbial heterogeneity and the expansion of some bacterial taxa, along with the secretion of bacterial toxins by Proteobacteria that alter nutrient absorption and activate cytokine-mediated systemic inflammation. A study by Qi et al. on a cohort of both older (>70 years) and younger patients demonstrated that with advancing age, there is an increase in serum zonulin and intestinal permeability, in correlation with systemic inflammation (TNFα, IL-6, HMGB1) and increased frailty [44].

During hospitalization, factors such as antibiotic use, dietary changes, and reduced mobility can worsen gut dysbiosis, which may affect the absorption of essential nutrients, such as proteins and amino acids, critical for regulating muscle metabolism [45].

Therefore, the interaction between oxidative stress and increased low-grade endotoxemia could create a vicious cycle that exacerbates sarcopenia, particularly in hospitalized older adult patients (Figure 2). Oxidative stress can alter the intestinal environment, increasing low-grade endotoxemia, which is a trigger for systemic inflammation and oxidative stress. This bidirectional relationship accelerates muscle degradation, making it more difficult to counteract the effects of sarcopenia (Figure 2).

The aim of our study was to quantify and measure the detrimental effect of hospitalization on sarcopenia in older adult patients and investigate the relationship with oxidative stress and low-grade endotoxemia.

This study quantified the impact of hospitalization on the loss of muscle mass and strength in hospitalized older adult patients; in fact, we observed a marked reduction in muscle tissue, equal to approximately 30% of the rectus femoris muscle, 28% of the diaphragm muscle, and 26% of the intercostal muscles between admission and discharge. In addition, a 25% reduction in muscle strength was observed by performing the hand grip test on admission and at discharge. Through the execution of Geriatric Assessment Scales, we intercepted how hospitalization affects older adult patients at 360 degrees; we observed a worsening of cognitive deficit, mood, nutritional status, sarcopenia, and disability at discharge compared to admission.

This study demonstrates increased serum levels of sNOX2-dp and H_2_O_2_ at discharge, indicating a consequent increase in oxidative stress after hospitalization. Furthermore, we observed an increase in intestinal permeability and thus low-grade endotoxemia with higher values of LSP and zonulin measured at discharge. This study also demonstrates that the worsening of sarcopenia observed during hospitalization is associated with an increase in serum levels of zonulin and LPS, which could, in turn, act as a trigger for the activation of NADPH oxidase type 2, leading to an increase after hospitalization. This condition may lead to chronic inflammation, which is also potentially detrimental to sarcopenia [16]. In agreement with our data, in fact, we confirm that there seems to be a correlation between increased age-related permeability of the intestinal barrier, the development of oxidative stress, systemic inflammation, and sarcopenia in hospitalized older adults [44].

It is thus necessary to counteract the widespread sarcopenic phenomenon in hospitalized older adult patients through multifactorial strategies. Potential approaches could include nutritional support (to provide adequate protein and antioxidants [46]), the use of prebiotic and probiotic supplements, early mobilization, and the encouragement of in-hospital physical exercise (to help maintain muscle mass and function).

Physiotherapy can be considered a potential approach to address this issue. Numerous studies in the literature have highlighted the beneficial role of physical exercise, particularly through the implementation of multimodal programs that combine aerobic resistance training, motor coordination, and balance exercises [47]. Additionally, nutritional supplementation should be considered, focusing on increased protein intake, adequate vitamin D levels, and the inclusion of natural antioxidants or probiotics [48].

The combination of targeted rehabilitation activities and nutritional supplementation, as demonstrated by the RCT conducted by Papadopoulou SK et al., which identified a synergistic effect between these two interventions, represents a promising avenue [49]. This integrated approach warrants further exploration, particularly within hospital settings [49].

This study has several limitations and implications. The limitations are due to the lack of oxidative stress analysis through muscle biopsies, which could have highlighted NOX2 activity and oxidative stress in the anatomical site. Furthermore, dysbiosis was not analyzed in these patients through gut microbiota analysis, nor were other NOX isoforms, such as isoform 4, which may be involved in muscle metabolism and catabolism. Zonulin is an indirect marker of gut permeability; therefore, we cannot exclude the possibility that serum LPS may also originate from other sources. Further studies are needed to support this hypothesis. An additional limitation of the study is the small sample size, which did not allow for an adjustment for confounding variables necessary to better define the comparison between hospitalized and outpatient participants as well as for hospitalized patients between the admission and discharge periods. The implications are manifold. This study aimed to establish a new standard of care that aims to counteract sarcopenia and its aggravating factors in older hospitalized patients. Reducing this phenomenon could lead to the development of new pre-hospitalization and in-hospital management protocols that can counteract the decrease in muscle mass and physical performance caused by the numerous factors described.

## 5. Conclusions

This study demonstrates that hospitalization of older adult patients leads to an increase in NOX2 blood levels, which is closely associated with a decline in muscle mass indices. The activation of NOX2 during hospitalization may be driven by increased low-grade endotoxemia, triggering elevated oxidative stress that contributes to muscle degeneration and, in severe cases, leads to sarcopenia.

Addressing this multifactorial condition requires a comprehensive approach, including nutritional support, physical exercise, and strategies to preserve gut health. Future targeted intervention studies should investigate the composition of the gut microbiota and evaluate the impact of altering it on mitigating sarcopenia. Additionally, implementing daily resistance training programs for hospitalized patients may help reduce both the prevalence and severity of sarcopenia, ultimately lowering associated complications and improving clinical outcomes for this vulnerable population.

## Figures and Tables

**Figure 1 antioxidants-14-00304-f001:**
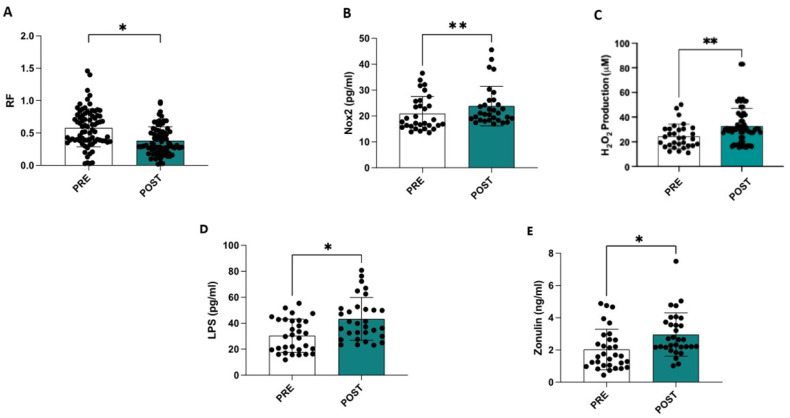
Hospitalization is associated with a reduction in the thickness of the rectus femoris muscle (**A**), an increase in serum NOX2 levels (**B**), an elevation in oxidative stress assessed by measuring serum H_2_O_2_ levels (**C**), and higher circulating levels of LPS (**D**) and zonulin (**E**). * *p* < 0.001; ** *p* < 0.01.

**Figure 2 antioxidants-14-00304-f002:**
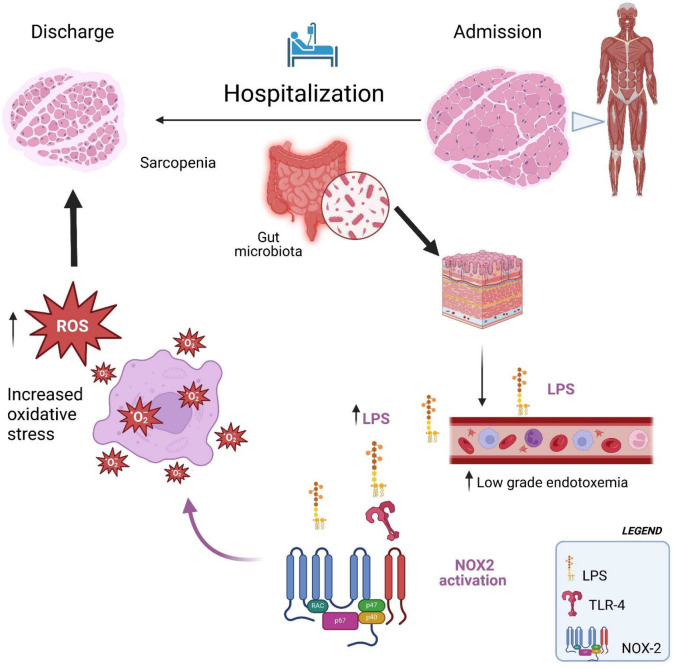
Hospitalization facilitates the translocation of LPS into the bloodstream, which in turn activates NOX2, leading to increased oxidative stress and exacerbating sarcopenia.

**Table 1 antioxidants-14-00304-t001:** Clinical characteristics of controls and patients.

Clinical Characteristics	Outpatients (n = 31)	Hospitalized Patients (n = 31)	*p*-Value
Age (years)	80 ± 4	82 ± 8	*p* = 0.210
Gender males/females	14/17	14/17	*p* = 1
NOX2 (pg/mL)	9.4 ± 5.6	20.9 ± 6.5	*p* < 0.001
Zonulin (ng/mL)	1.5 ± 0.43	2.06 ± 1.23	*p* = 0.007
LPS (pg/mL)	11.2 ± 3.2	30.4 ± 12.6	*p* < 0.001
H_2_O_2_ µM	9.3 ± 3.04	24.4 ± 9.8	*p* < 0.001
MMSE	27.7 ± 1.3	19.3 ± 9.9	*p* < 0.001
Barthel	89.8 ± 5.6	45.6 ± 34.4	*p* < 0.001
Arterial Hypertension yes/no	23/8	22/9	*p* = 0.745
Diabetes Mellitus yes/no	3/28	14/17	*p* < 0.001
Dyslipidaemia yes/no	13/18	21/10	*p* = 0.020
Heart Failure yes/no	0/31	10/21	*p* < 0.001
Former Smokers yes/no	0/31	13/18	*p* < 0.001
Dementia yes/no	0/31	15/16	*p* < 0.001
BMI	24.6 ± 2.4	21.8 ± 3.17	*p* < 0.001
Days of hospitalization	-	14.4 ± 8.1	*p* < 0.001
Antibiotic therapy yes/no	0/31	19/12	*p* < 0.001
Statin therapy yes/no	13/18	20/11	*p* < 0.001
Anticoagulant therapy yes/no	0/31	20/11	*p* < 0.001
Antiplatelet therapy yes/no	0/31	20/11	*p* < 0.001
Pneumonia yes/no	0/31	19/12	*p* < 0.001
Anemia yes/no	0/31	14/17	*p* < 0.001
Chronic Ischaemic Heart disease yes/no	0/31	17/14	*p* < 0.001
Previous Venous Thromboembolism yes/no	0/31	0/31	*p* < 0.001
Triglycerides mg/dL	99 ± 37	73.9 ± 19	*p* < 0.001
Total cholesterol mg/dL	131 ± 44.6	144 ± 27	*p* < 0.001

**Table 2 antioxidants-14-00304-t002:** Characteristics of patients at entry and discharge.

Assessment	Admission	Discharge	*p*-Value
Barthel Scale	45.6 ± 34.4	33.8 ± 31.6	*p* < 0.001
Rectus femoris (RF) muscle thickness cm	0.5 ± 0.29	0.3 ± 0.31	*p* < 0.001
MMSE	19.3 ± 9.9	17.7 ± 10	*p* < 0.001
GDS	6.6 ±2.7	9.4 ± 2.6	*p* < 0.001
CIRS	18.4 ± 10.4	22 ± 10.2	*p* < 0.001
MNA	17.1 ± 3.6	13.1 ± 2.3	*p* < 0.001
Hand grip test	13.4 ± 10.8	10.0 ± 8.2	*p* < 0.001
SARC-F	5.01 ± 1.16	7.6 ± 0.78	*p* < 0.001
Intercostal muscle thickness cm	0.22 ± 0.08	0.16 ± 0.06	*p* < 0.001
Diaphragmatic muscle thickness cm	0.19 ± 0.06	0.14 ± 0.04	*p* < 0.001
sNOX2-dp (pg/mL)	20.9 ± 6.5	23.8 ± 7.5	*p* = 0.004
H_2_O_2_ µM	24.4 ± 9.8	32.8 ± 14.5	*p* = 0.01
LPS pg/ml	30.4 ± 12.6	43.3 ± 16.35	*p* < 0.001
Zonulin (ng/mL)	2.06 ± 1.23	2.95 ± 1.33	*p* < 0.001
Leg circumference cm	36.06 ± 5.3	31.48 ± 2.5	*p* < 0.001
Arm circumference cm	24.90 ± 2.9	21.29 ± 1.9	*p* < 0.001

## Data Availability

Data are available on request to the corresponding author.

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
