# Peer review of "Impact of Hospitalization on Sarcopenia, NADPH-Oxidase 2, Oxidative Stress, and Low-Grade Endotoxemia in Elderly Patients"

_antioxidants, 2025, doi:10.3390/antiox14030304_

Round 1
Reviewer 1 Report
Deat authors,
your work "Impact of Hospitalization on Sarcopenia, NADPH-Oxidase, Oxidative Stress, and Low-Grade Endotoxemia in Elderly Patients."with the aim of "investigate the impact of hospitalization on muscle mass in elderly patients, focusing on the roles of oxidative stress and gut-derived low-grade endotoxemia." is interesting but the manuscript presented is confusing and it is hard to read. Here are my suggestions:
a)Abstract is too long;
b)Introduction should contain the study goals here; and the sentences should be rewrited in a more precise direction to your study aim.
c) methods: format according to journal standarts; add exclusion criteria
d) discussion: Format figure 2 and resume title.
e)conclusion: Be more precise and add a more direct conclusion according to your results.
all in major comments
Reviewer 2 Report
The manuscript ”Impact of Hospitalization on Sarcopenia, NADPH-Oxidase 2, Oxidative Stress, and Low-Grade Endotoxemia in Elderly Patients” presents a compelling investigation into the relationship between hospitalization, sarcopenia, oxidative stress, and endotoxemia in elderly patients and is an important contribution to the field of geriatric medicine and oxidative stress. The study emphasizes the need for multifactorial interventions, such as nutritional support and physical rehabilitation, to mitigate muscle loss in hospitalized elderly patients. The correlation analyses linking NOX2 with muscle atrophy indices provide compelling evidence supporting the role of oxidative stress in hospitalization-induced sarcopenia. Despite these strengths, several aspects require improvement. The study's sample size is relatively small (n=31 per group), limiting the generalizability of the findings. The discussion section, while insightful, could benefit from a more structured progression from key findings to mechanistic interpretations, ensuring that readers fully grasp the study's implications. Lastly, statistical adjustments for confounders such as medication use, BMI, and pre-existing conditions would enhance the robustness of the reported associations.
1. Abstract (Lines 29-57): The results section should explicitly mention effect sizes and confidence intervals to provide a more quantitative overview. Including standard deviations for key biomarkers would enhance transparency.
2. Methodology (Lines 119-144): The study does not clearly state whether muscle ultrasound assessments were blinded. Clarifying if the ultrasound operator was blinded to patient status would improve methodological rigor.
3. Statistical Analysis (Lines 214-220): The statistical methods section should specify if corrections for multiple comparisons were applied. This is crucial given the multiple biomarkers analyzed.
4. Demographics: The presence of confounding factors (e.g., diabetes, dementia) between groups should be statistically adjusted otherwise, the comparisons may not be valid.
5. Terminology: The manuscript interchangeably uses "sarcopenia" and "muscle degradation." Standardizing terminology throughout the text would improve readability and scientific precision.
6. Figures should include legends with clear descriptions of the statistical comparisons (e.g., specifying whether the p-values refer to baseline vs. discharge or between groups).
7. Missing Dietary Assessment: Nutrition is a key factor in sarcopenia, yet dietary intake was not assessed. If dietary data were unavailable, this limitation should be explicitly acknowledged.
8. The sample size is relatively small (n=31 per group). The generalizability of findings should be discussed, including a power analysis to justify sample selection.
9. The regression models lack adjustments for key confounders such as BMI, medication use, or physical activity. These should be incorporated to improve the validity of the conclusions.
10. (Lines 272-280): While NOX2 involvement is emphasized, alternative sources of oxidative stress (e.g., mitochondrial dysfunction) are not discussed.
11. (Lines 298-303): The study discusses endotoxemia but does not present direct gut microbiome data. While relevant references are cited, emphasizing the limitations of assuming dysbiosis from zonulin alone is necessary.
12. (Lines 260-335): The discussion should begin with a concise summary of key findings before exploring mechanistic interpretations. The discussion begins like a conclusion: This study demonstrates the significant impact of hospitalization on muscle reduction in elderly hospitalized patients.
13. Some tables lack horizontal borders, making it difficult to track rows (e.g., Table 1). Standardizing table formatting according to journal guidelines would improve readability.
14. The conclusion should specify concrete recommendations for future research (e.g., intervention studies, microbiome analysis) rather than broadly suggesting "multifactorial approaches." Specificity would strengthen the final remarks.
Round 2
Reviewer 1 Report
Dear authors,
you have attended all my suggestions.
Dear authors,
you have attended all my suggestions.
Reviewer 2 Report
The authors satisfactorily responded to all 14 previous comments and suggestions point-by-point. For some points, they acknowledged the limitations accordingly. Based on these facts, I conclude with acceptance.
-